Changes in tooth size of Otaria byronia: an indicator of density-dependent effects?

Sosa Drouville Ailin 1 ailinsosad@gmail.com
Heredia Federico 2
Coscarella Mariano A. 2 3
http://orcid.org/0000-0001-9216-7817 Crespo Enrique 2 3
http://orcid.org/0000-0002-1418-4205 Grandi María Florencia 2
1 Instituto de Biología de Organismos Marinos (Consejo Nacional de Investigaciones Científicas y Técnicas) , Puerto Madryn , Argentina
2 Centro para el Estudio de Sistemas Marinos (Consejo Nacional de Investigaciones Científicas y Técnicas) , Puerto Madryn , Argentina
3 Universidad Nacional de la Patagonia San Juan Bosco , Puerto Madryn , Argentina
Manjarrez Javier
Electronic publication date: 2025 Mar 7
Publication date: 2025
Volume: 13
Electronic Location ID: e18963
Received 2024 Nov 18; Accepted 2025 Jan 20
Copyright: © 2025 Sosa Drouville et al.
Copyright year: 2025
Copyright holder: Sosa Drouville et al.
License: This is an open access article distributed under the terms of the Creative Commons Attribution License, which permits unrestricted use, distribution, reproduction and adaptation in any medium and for any purpose provided that it is properly attributed. For attribution, the original author(s), title, publication source (PeerJ) and either DOI or URL of the article must be cited.
License URL: https://creativecommons.org/licenses/by/4.0/

Keywords: Density-dependence, Pinnipeds, Teeth, South American sea lion, Otaria byronia

Funding: Agencia Nacional de Promoción CientÌfica y Tecnológica PICTs 2110, 33934, 11679, 2063 AmnÈville zoo France (2004–2020) Mohamed Bin Zayed species conservation fund (2011–2014) Yaqupacha, Heidelberg zoo (2012) BBVA BIOCON_08, 2009-2012 CONICET PIP 0111/2016 The Explorers Club This research was funded by the Agencia Nacional de Promoción CientÌfica y Tecnológica (PICTs 2110, 33934, 11679, 2063), AmnÈville zoo France (2004–2020), the Mohamed Bin Zayed species conservation fund (2011–2014), Yaqupacha, Heidelberg zoo (2012), BBVA (BIOCON_08, 2009-2012) and CONICET (PIP 0111/2016), granted to Enrique Alberto Crespo. Additionally, by The Explorers Club (granted to Ailin Sosa Drouville). The funders had no role in study design, data collection and analysis, decision to publish, or preparation of the manuscript.

==============================
Teeth provide valuable information about an individual’s life cycle and serve as a powerful tool for visualizing population-level changes associated with density-dependent processes. In pinnipeds, teeth are used to estimate the age of individuals based on the count of growth layer groups (GLG) in the dentine. In this study, we analyzed changes in tooth size and GLG widths in the canines of Otaria byronia throughout the past 100 years, linking these to fluctuations in population abundance. A total of 76 male individuals from Patagonia were analyzed, classified into two periods: harvest and postharvest. The length and diameter of each tooth were measured prior to sagittal sectioning. Only for the postharvest period body length was recorded. Longitudinal sections of the upper canine were prepared, age was determined and the width of each GLG was measured in the resulting half-tooth. Results indicated a positive correlation (F = 62.90; p < 0.001; n = 50; r2 = 0.59) between body length and tooth length in postharvest individuals, suggesting that tooth growth is a reliable indicator of body growth. Individuals from the harvest period had narrower GLGs (t = 3.75; p < 0.001) and smaller tooth size (t = 3.48; p < 0.001) compared to those from the postharvest period. These results indicate that somatic growth of individuals may vary with population numbers and resources available. Also, hard structures like teeth are excellent tools for visualizing density-dependence effects.

Introduction

Density-dependence is a mechanism that occurs in K strategist species and is noted or measured when population abundance approaches carrying capacity (McLaren & Smith, 1985). The changes caused by density-dependence occur dramatically and can be seen in a relatively short time (Fowler, 1981; Lima, 1995). Density-dependent regulation can generate physiological or behavioral changes in individuals within a population (Fowler, 1990; Valenzuela-Toro et al., 2023). In particular, hard structures like teeth provide information about an individual’s life cycle and are powerful tools for visualizing population-level changes associated with density-dependent processes.

Age determination of individuals is one of the most important parameters for studying the population dynamics of mammals (McLaren & Smith, 1985; Scheffer & Myrick, 1980). In the past, several methods have been used to estimate the approximate age of individuals, such as body length, lens weight, cranial sutures, tooth wear, and corpora albicantia count (McLaren & Smith, 1985). However, none of these techniques provided precise information on the age of an individual (Scheffer & Myrick, 1980). Since the 1950s, pinniped teeth have been used to estimate ages by counting growth layer groups (GLGs) found in the dentine and/or the cement (Scheffer, 1950). This technique is mostly used in marine mammals as it provides more accurate information on the chronological age of the individual (Crespo, 1988; Laws, 1952, 1953; Loza et al., 2016; Read, Hohn & Lockyer, 2018; Scheffer, 1955). Particularly in otariids, the GLGs are better observed in the canine. These annual growth layers are present in an ever-growing dentition.

In pinnipeds, it is generally assumed that one GLG corresponds to the amount of tissue accumulated during a year of life. In tooth-thin sections observed with transmitted light, a GLG is composed of a thin, clear band and a broader, opaque band (valley and ridge, respectively, in acid-etched sections of half-tooth) (Crespo, 1988; Crespo et al., 1994; Laws, 1952, 1953, 1962; Scheffer, 1955). The opaque bands correspond to the feeding period, and the light bands correspond to the fasting season usually associated with the reproductive season (Crespo, 1988). The deposition of a GLG can be modified by physiological events such as pregnancy, lactation, weaning, fasting, moulting, and sexual maturity (Bengtson, 1988; Boyd & Roberts, 1993; Mansfield, 1991), or by extreme climatic conditions (Heredia et al., 2021; Dellabianca et al., 2012; Wittmann et al., 2016).

The dentine of pinnipeds, like that of other mammals, is composed of 35% organic components, mainly collagen fibers and mucopolysaccharides, and 65% inorganic components, including hydroxyapatite and small amounts of zinc, strontium, fluorine, magnesium, manganese, lead, iron and tin (Klevezal, Mina & Oreshkin, 1996). Dentine is formed by the activity of cells called odontoblasts, which are located in the wall of the pulp cavity (Klevezal, Mina & Oreshkin, 1996). Particularly, dentine presents a high sensitivity to diet changes, and the mineralization process of GLGs depends on the contribution of vitamins and minerals ingested with food (Klevezal, Mina & Oreshkin, 1996; Laws, 1962). Consequently, the physiological mechanisms of mineralization can be affected by a nutritional deficit, leading to sparse deposition of minerals that structure the GLG (Boyd & Roberts, 1993; Hanson et al., 2009; Klevezal, Mina & Oreshkin, 1996; Knox et al., 2014; Wittmann et al., 2016). Since GLGs are deposited from the pulp cavity, it is important to consider the normal progressive decrease in the width of the GLG with the age of the animal before relating this information to food availability or growth (Boyd & Roberts, 1993). Therefore, the various events throughout an individual’s life cycle are reflected in the dentine deposits on their teeth. This could generate patterns in the tooth growth that could reflect the life history of the individual (Boyd & Roberts, 1993; Newsome et al., 2006, 2007).

The South American sea lion (SASL, Otaria byronia) population from Patagonia has experienced a drastic reduction in size over a relatively short period, and then began a slow recovery after sealing ended (Crespo & Pedraza, 1991; Dans et al., 2004; Grandi, Dans & Crespo, 2015; Reyes, Crespo & Szapkievich, 1999; Romero et al., 2017). In Argentina, large scale commercial sealing of SASL began in 1917 and ended in 1962 (Romero et al., 2017). Skin and blubber were used for leather, fur and oil production, and the remains of the animals were discarded on the coast, generating ossuaries near the factories (Bastida, 1963; Carrara, 1952; Crespo & Pedraza, 1991). Península Valdés (in northern Patagonia), Tierra del Fuego and Malvinas Islands supported the most heavily exploited stocks of the Atlantic Ocean (Romero et al., 2017; Baylis et al., 2015). The northern and central Patagonian population dropped drastically from an estimated 440,000 individuals in the preharvest period to 20,000 individuals in a few years, a reduction of more than 90% of its original abundance (Romero et al., 2017). After the harvest ceased, the population reached its minimum abundance and then began to recover (Romero et al., 2017). Romero et al. (2017) observed that populations of SASL have a nonlinear relationship with density, assuming an “overcrowding” or compensatory density-dependent process that affects the population growth rate at high densities. These changes in abundance over time provide a favorable scenario for testing possible changes related to density-dependent phenomena. Before the commercial harvest, SALS abundance was probably in equilibrium with the per capita food availability in the environment, resulting in individuals of a certain body size. At the end of the harvest, population density was so low that individuals likely had greater per capita food availability that ultimately would lead to a larger body size.

The variation in individual size could also be related to the reaction norm of the genotype. Norm of reaction represents the range of phenotypic variation produced by a genotype in response to environmental variation (Woltereck, 1909). Studying reaction norms is important for understanding various aspects of phenotypic evolution (Bhumika & Singh, 2019). Phenotypic plasticity is defined as the ability of a particular genotype to produce more than one phenotype in response to changing environmental conditions such as temperature, population density, nutrition, etc. (Yang & Pospisilik, 2019). Phenotypic plasticity provides species with the ability to facilitate adaptive changes and increase phenotypic diversity, thereby enabling them to better cope with environmental changes (Yang & Pospisilik, 2019). In this context, it is expected that individual growth will be reflected in teeth growth patterns, which could be modified over time due to density-dependent effects and the norm of reaction of the genotype.

Therefore, the objective of this work was to analyze potential changes in the upper canines of male Otaria byronia, related to changes in population abundance over the past 100 years.

Materials and Methods

Sample and study area

A total of 76 upper canine teeth of male SASL from northern and central Patagonia were analyzed (Fig. 1). The sample belongs to the Osteological Marine Mammal Scientific Collection of CESIMAR—CONICET, Argentina (Table S1).

Figure 1 Study area indicating sampling locations of South American sea lions in northern and central Patagonia, Argentina (●) and Punta Norte ossuary (▲).

Individuals were classified into periods according to their time of death: harvest (1917–1962) and postharvest (1963–2017). The individuals from the harvest period (n = 26) corresponded to the sealing time when population abundance was high and presumably at carrying capacity. These individuals were obtained from the ossuary at Punta Norte (42°04′S, 63°45′W), Península Valdés, Argentina (Fig. 1). The postharvest period includes individuals (n = 50) found dead on the coasts or incidentally caught in commercial trawl fisheries in northern and central Patagonia (Fig. 1). Only sub-adult and adult individuals (i.e., older than 4 years, are sexually mature, Grandi et al. (2010)) were selected to avoid differences related to ontogeny.

Data collection and tooth preparation

The length (L) and diameter (D) of each tooth were measured using a digital caliper (Mitutoyo, minimum value 0.01 mm), before sagittal sectioning. For the postharvest period, body length (standard length (LS)) was recorded using a measuring tape (minimum measurement 1 mm). Subsequently, upper canines were sagittally sectioned through the center of the pulp cavity with a handsaw. The best half-tooth was selected, polished, and etched in 5% nitric acid. Each half-tooth was then rinsed, dried at room temperature, and rubbed with acetone to enhance GLG contrast (Fig. 2) (Crespo et al., 1994).

Figure 2 Sagittal section of the upper canine of a South American sea lion male.

It is noted the pulp cavity, dentine, cement and enamel. In the dentine the GLGs are marked with graphite.

The inner surface of the half-teeth was photographed using a Cannon Rebel camera. Two observers independently counted the number of GLGs on each half-tooth in different reading sessions, and final age assignments were based on consensus (Table S1). Then GLG width was measured in the dentine using Leica Application Suite V3.4.0 software, which allowed for plotting lines on the edge of each GLG and measuring width considering the scale of the photograph. Measurements were taken on the polished surface of the most concave side of the half-tooth (Fig. 2), from the neonatal line (i.e., the line laid at birth) to the pulp cavity, in a staggered manner (Fig. 2). As GLG width varies throughout its entire path, the measurements were taken in the most stable width area (i.e., the central area of the tooth). Measurements of the first GLG were excluded from the sample, since they presented high variability.

Data analysis

To study the relationship between tooth length (L) and individual size, linear regression was performed to analyze whether tooth size is a good proxy of body size. This was done using postharvest individuals, as body length data from the harvest period individuals were unavailable. T-Student analyses were conducted to evaluate differences in tooth length (L) and diameter (D) between the harvest and postharvest periods.

Generalized linear mixed models with a normal error distribution were used to assess whether there were differences in the width of the growth layers (GLGW) between periods. The response variable, GLGW, was modeled using two predictor variables: the period as a categorical variable (harvest and postharvest) and n°GLG as a continuous variable (which is the number of each growing layer). Individual id was used as a random covariable. Two of the most suited autocorrelation structures were used to model the temporal dependence of the response variable within each tooth. The gls function from the “MASS” library (Venables & Ripley, 2002) and the lme function of the “nlme” v. 3.1-127 package (Pinheiro & Bates, 2000) were implemented with R (R Core Team, 2017). The modelling procedures followed Zuur et al. (2009), and models were selected using the Akaike information criterion (AIC).

Results

The body length and tooth length in individuals from the postharvest period showed a significant linear regression (F = 62.90; p < 0.001; n = 50; r2 = 0.59), suggesting that tooth growth is a good indicator of body growth (Fig. 3). Additionally, there were differences in tooth length (L) and diameter (D) between harvest and postharvest period, with teeth from the harvest period being significantly shorter (t = 3.48; p < 0.001; Fig. 4) and thinner (t = 3.75; p < 0.001; Fig. 4) compared to those from the postharvest period.

Figure 3 Scatter plot and linear regression between body length and tooth length of SASL males during the Post-harvest period.

Figure 4 Left: Boxplot of tooth length of the SASL males of both periods.

Right: Boxplot of tooth diameter of the SASL males of both periods. (Red) Harvest period and (white) Post-harvest period.

The best models are presented in terms of ΔAIC (Table 1), and as a rule of thumb, values that are less than two should be given consideration for the selected model (Burnham & Anderson, 2004). The Akaike information criterion (AIC) favours Model 4, as shown in Table 1. Model 4 includes the predictor variables: period (harvest, post-harvest), n°GLG, individual as a random effect, and an Auto Regression Model 1 (AR1) temporal autocorrelation structure (Table 1). All variables were significant.

Table 1 Summary of the generalized linear mixed models used.

Models	Models	ΔAIC	df	
M4	Lme (GLGW ~ n°GLG * period, random = ~1|id, na.action=na.omit, correlation=corCompSymm (form = ~1|id/glb))	7.5	7	
M2	Lme (GLGW ~ n°GLG * period, random = ~1|id, correlation=corAR1 (form = ~1| id/glb))	5.1	7	
M3	Lme (GLGW ~ n°GLG * period, random = ~1|id, na.action=na.omit, correlation=corCompSymm (form = ~1|id/glb))	7.5	7	
M1	Lme (GLGW ~ n°GLG * period, random = ~1|id)	108.4	6	
Note:

The structure for each model, ΔAIC and df values are shown.

The model results indicated that individuals from the harvest period had smaller GLG widths compared to those from the postharvest period (Fig. 5). Additionally, the model’s slopes are equal, showing that the decrease in the width of the growth layers was the same in both periods.

Figure 5 Relationship between GLGW and the N° GLG in both time periods (with interval of 95%).

(Red) Harvest and (grey) Post-harvest.

Discussion

This study demonstrates that the growth of canine teeth is a good indicator of body growth of individuals. The results suggest the existence of a density-dependent response in tooth growth of male SASLs of Patagonia, Argentina. GLGs were found to be thinner in individuals from the harvest period, compared to post-harvest individuals, likely due to higher population density in sealing time. Additionally, teeth from the harvest period were thinner and shorter than those from the post-harvest period, indicating changes in somatic growth over time. In turn, the results also suggest that a lesser amount of dentine (i.e., narrower GLG) was deposited in each calendar year in individuals from the harvest period compared to post-harvest individuals.

Changes in population abundance have a significant impact on individual growth. These changes are often related to density-dependent processes that produce physiological or behavioral changes in individuals of a population (Fowler, 1990). Therefore, differences in GLG width and tooth size between the two time periods are likely related to changes in O. flavescens population abundance. Before commercial harvesting, the population of northern and central Patagonia was estimated at 440,000 individuals (Romero et al., 2017). At its height, intraspecific competition would have been the highest, leading to a lower per capita intake, likely investing more energy in searching for food, consuming a smaller amount of prey per capita and/or prey of lower nutritional quality. This results in lesser dentine deposits in GLGs and could be ultimately reflected in lower somatic growth (Sosa Drouville et al., 2021). On the contrary, when the population declined, the intraspecific competition was lower, leading to an increase in dentine deposits thickness and larger somatic growth.

Dietary changes caused by density-dependent factors may be reflected in the physical condition of individuals and their somatic growth (Trites & Bigg, 1992; Sosa Drouville et al., 2021). The results obtained are consistent with findings in other studies (Boyd & Roberts, 1993; Etnier, 2004; Hanson et al., 2009; Sosa Drouville et al., 2021; Scheffer, 1955). For example, the northern fur seal, Callorhinus ursinus, shows a decrease in body size with increased population size (Etnier, 2004; Scheffer, 1955). In the Arctocephalus gazella population from South Georgia Islands, GLG width decreased with increased population density (Boyd & Roberts, 1993). A decline in annular tooth growth was observed in male Antarctic fur seal, A. gazella, from South Georgia as a consequence of rapid population growth (Hanson et al., 2009).

In pinnipeds, there are marked intra-specific differences in feeding patterns. Diving skills, dive duration, dive depth, distance to shore, and swimming speed increase with age and are acquired progressively throughout their lives (Bekkby & Bjørge, 2000; Costa, 1991; Chilvers et al., 2005, 2006; Horning & Trillmich, 1997). Hence, an increase in population abundance likely increases prey consumption near colonies, leading to higher intraspecific competition (Drago et al., 2010, 2011). Females are particularly affected due to the constrains of raising pups, especially during the early lactation period when foraging trips are restricted in distance and duration by the fasting ability of pups (Drago et al., 2010; Riet-Sapriza et al., 2013). Suboptimal maternal nutrition may result in lower offspring somatic growth and smaller GLG width during lactation. Pups likely spend their early years feeding in areas surrounding the colony with short foraging trips. Drago et al. (2011) found that O. flavescens pups grew slower in more abundant colonies than in smaller ones, and this may be due to lower per capita food availability for females. Therefore, if food is a limiting factor in the feeding areas near the colony, there may be a nutritional deficit that could be reflected in tooth growth and GLG composition by poor mineral deposition. In this study, we found that males of O. flavescens exhibit the same pattern that would be expected for females that feed near the colonies, even though the males feed further away (Koen Alonso et al., 2000; Campagna et al., 2001). This could be the result of an onset effect due to insufficient nutrition during the lactating period, which affects the male’s development throughout its life, higher competition in the feeding grounds despite being further away, or a combination of both (Sosa Drouville, 2023).

Differences found in tooth size between the two periods may also be related to the norm of reaction of the genotype, producing different phenotypes under a gradient of environmental conditions such as population abundance, diet, and behavioral changes (Forsman, 2015). Genes that code for tooth growth can have different phenotypic outputs that can be influenced by environmental conditions. Optimal environmental conditions yield maximum growth values, while suboptimal conditions result in minimum value (Woltereck, 1909). Therefore, we propose that the difference in tooth size observed in the present work could likely be due to the intense intraspecific competition generated by the high population density during the harvest period, preventing teeth from reaching their maximum growth potential determined by the genotype of the species.

Tooth structure can also provide powerful and relevant information about individual and population life history. The upper canines of SASL males are large and robust, with clearly visible GLGs in the dentine. Pinniped growth patterns reflect species characteristics and environmental features (McLaren & Smith, 1985). Climate patterns can have a great impact on the somatic growth of individuals and may affect the availability of resources in the environment (Heredia et al., 2021; Sielfeld, Barraza & Amado, 2018; Sprogis et al., 2018). For example, SASL population in Chile alters their diet during El Niño-Southern Oscillation (ENSO) events, leading to nutritional stress and affecting somatic growth, survival rates, birth rates, and increased susceptibility to disease (Sielfeld, Barraza & Amado, 2018). Another example in Patagonia, SASL female GLG width was influenced by the Southern Annular Mode (SAM) climate pattern (Heredia et al., 2021). Regarding SASL males from Patagonia, there was no alteration in GLGs growth associated with SAM or ENSO (Heredia et al., 2021).

On the other hand, changes in the diet of a species or changes in prey availability are often associated with industrial fishing and could generate an impact in the somatic growth (Crespo et al., 1997; Dans et al., 2003). In the Argentine Sea, fishing activity targeting hake (Merluccius hubbsi), shrimp (Pleoticus muelleri) and squid (Illex argentinus) increased considerably since the 1970s (Bertolotti et al., 2001; Bezzi & Dato, 1995; Brunetti, 1990), leading to the decline of several stocks due to over-exploitation by the 2000s (Cordo, 2004). These species are the main prey in the diet of SASL (Koen Alonso et al., 2000; Sosa Drouville, 2023), making the fishing industry a crucial factor affecting food availability from the post-harvest period.

This study covers over 100 years of the life history of the northern and central Patagonian sea lion population, confirming that hard structures like teeth are excellent tools for visualizing the effects of density-dependence. Teeth degrade slowly, making them ideal for studying long-term changes in individual and population life histories.

Supplemental Information

Supplemental Information 1 Individual Id, Time period and age of South American sea lions used for the analyses.

Supplemental Information 2 Raw data 1 used in statistical analysis.

Supplemental Information 3 Raw data 2 used in statistical analysis.

We wish to express our gratitude to the staff of the Administración de Parques Nacionales, Dirección de Fauna y Flora Silvestre of Chubut and Secretaría de Ambiente y Cambio Climático of Río Negro, for permits to conduct research in protected areas.

Additional Information and Declarations

Competing Interests

Mariano Coscarella is an Academic Editor for PeerJ.

Author Contributions

Ailin Sosa Drouville conceived and designed the experiments, performed the experiments, analyzed the data, prepared figures and/or tables, authored or reviewed drafts of the article, and approved the final draft.

Federico Heredia conceived and designed the experiments, performed the experiments, analyzed the data, authored or reviewed drafts of the article, and approved the final draft.

Mariano A. Coscarella conceived and designed the experiments, analyzed the data, authored or reviewed drafts of the article, and approved the final draft.

Enrique Crespo conceived and designed the experiments, authored or reviewed drafts of the article, and approved the final draft.

María Florencia Grandi conceived and designed the experiments, authored or reviewed drafts of the article, and approved the final draft.

Data Availability

The following information was supplied regarding data availability:

The raw data are available in the Supplemental Files.

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
