# Peer review of "Changes in tooth size of Otaria byronia: an indicator of density-dependent effects?"

_PeerJ, doi:10.7717/peerj.18963_

## Round 0.1 · original submission · Minor Revisions

After reviewing this revised version of your manuscript, I see that the main comments suggested by the reviewers have been included. However, there are still some details that need to be clarified before having a final version that can be published.

Reviewer 1 ·

Basic reporting

no comment

Experimental design

no comment

Validity of the findings

no comment

Additional comments

This manuscript provides new data regarding increase in the size of the upper canine in South American sea lions within the period of a hundert years, and supports the idea based on findings from other pinnipeds and other mammals that tooth size may reflect changes in population abundance. I have found no problems of importance within this manuscript except for the scientific name of the animals examined. While there has been controversy surrounding the scientific specific name for South American sea lions, which results in two names in current use, Otaria flavescens and Otaria byronia, the former name is apparently a junior synonym of the latter (see Brunner 2004, https://doi.org/10.1017/S147720000300121X) and, therefore, the latter should be used according to the International Code of Zoological Nomenclature.
Minor problems:
1. Line 25: delete “that” from between “indicated” and “a positive”.
2. Line 63: insert a comma between “components” and “mainly”.
3. Add the reference to “Campagna et al. 2001” (line 226).
4. Line 230: use “norm” (not capitalised).
5. Lines 309, 421, 452, 460 and 495: Cabrera 1940, Lucero et al. 2019, Rice 1998, Rodriguez & Bastida 1993 and Vaz Ferreira 1982 are not cited elsewhere in the manuscript.

·

Basic reporting

The article has clear English, good referencing, structure, and a clear hypothesis. Raw data is provided, but it might be beneficial to also provide the R code used to analyze the data.

Experimental design

No comment

Validity of the findings

No comment

Additional comments

To the authors,

This is an excellent manuscript which is clear, well written, and a good contribution. I only have a few minor comments (below, and annotations to the PDF):

- In the Introduction (lines 35-41) it would be good to cite the following paper, which investigated density-dependent shifts in California Sea lions

Valenzuela-Toro, A. M., Costa, D. P., Mehta, R., Pyenson, N. D., & Koch, P. L. (2023). Unexpected decadal density-dependent shifts in California sea lion size, morphology, and foraging niche. Current Biology, 33(10), 2111-2119.

- When describing the tooth growth mechanisms on lines 47-51, and lines 52-56, it might be good to mention (if applicable): a) that in otariids, it is the canine that displays the potential annual growth layers, and b) that these annual growth layers are only present in ever-growing dentition.

- In lines 78-98, it is mentioned several times that sealing occurred over a short period, and at some point sealing ended. But no actual dates (specifically, the year) are mentioned. Please provide the years if known in this paragraph, to better contextualize the study.

- The final line (263-264) states that teeth degrade slowly. However, they still degrade, either by chemical abrasion from feeding, attrition, or biting during conflict. Could this potentially affect the observed growth layers if the enamel is worn away exposing the dentine? For the methodology, did you select unworn specimens in the first instance?

I look forward to reading the final published manuscript.

From Dr James Rule

---

## Round 0.2 · accepted · Accept

After reviewing this revised version of your manuscript, I see that the main comments suggested by the reviewers have been included. Therefore, I am satisfied with the current version and consider it ready for publication.